# Wireless Sensing of Lower Lip and Thumb-Index Finger ‘Ramp-and-Hold’ Isometric Force Dynamics in a Small Cohort of Unilateral MCA Stroke: Discussion of Preliminary Findings

**DOI:** 10.3390/s20041221

**Published:** 2020-02-23

**Authors:** Steven Barlow, Rebecca Custead, Jaehoon Lee, Mohsen Hozan, Jacob Greenwood

**Affiliations:** 1Department of Special Education and Communication Disorders, University of Nebraska, 141 Barkley Memorial Center, Lincoln, NE 68583-0738, USA; rcustead@gmail.com (R.C.); hozan@huskers.unl.edu (M.H.); jacob.greenwood@huskers.unl.edu (J.G.); 2Department of Biological Systems Engineering, University of Nebraska, 230 L.W. Chase Hall, Lincoln, NE 68583-0726, USA; 3Center for Brain-Biology-Behavior, University of Nebraska, C89 East Stadium, Lincoln, NE 68588-0156, USA; 4Department of Educational Psychology & Leadership, Texas Tech University, PO Box 41071, Lubbock, TX 79409, USA; jaehoon.lee@ttu.edu

**Keywords:** lower lip, thumb-index finger, isometric force, wireless sensing, middle cerebral artery, infarct

## Abstract

Automated wireless sensing of force dynamics during a visuomotor control task was used to rapidly assess residual motor function during finger pinch (right and left hand) and lower lip compression in a cohort of seven adult males with chronic, unilateral middle cerebral artery (MCA) stroke with infarct confirmed by anatomic magnetic resonance imaging (MRI). A matched cohort of 25 neurotypical adult males served as controls. Dependent variables were extracted from digitized records of ‘ramp-and-hold’ isometric contractions to target levels (0.25, 0.5, 1, and 2 Newtons) presented in a randomized block design; and included force reaction time, peak force, and dF/dt_max_ associated with force recruitment, and end-point accuracy and variability metrics during the contraction hold-phase (mean, SD, criterion percentage ‘on-target’). Maximum voluntary contraction force (MVCF) was also assessed to establish the force operating range. Results based on linear mixed modeling (LMM, adjusted for age and handedness) revealed significant patterns of dissolution in fine force regulation among MCA stroke participants, especially for the contralesional thumb-index finger followed by the ipsilesional digits, and the lower lip. For example, the contralesional thumb-index finger manifest increased reaction time, and greater overshoot in peak force during recruitment compared to controls. Impaired force regulation among MCA stroke participants during the contraction hold-phase was associated with significant increases in force SD, and dramatic reduction in the ability to regulate force output within prescribed target force window (±5% of target). Impaired force regulation during contraction hold-phase was greatest in the contralesional hand muscle group, followed by significant dissolution in ipsilateral digits, with smaller effects found for lower lip. These changes in fine force dynamics were accompanied by large reductions in the MVCF with the LMM marginal means for contralesional and ipsilesional pinch forces at just 34.77% (15.93 N vs. 45.82 N) and 66.45% (27.23 N vs. 40.98 N) of control performance, respectively. Biomechanical measures of fine force and MVCF performance in adult stroke survivors provide valuable information on the profile of residual motor function which can help inform clinical treatment strategies and quantitatively monitor the efficacy of rehabilitation or neuroprotection strategies.

## 1. Introduction

Ischemic brain damage associated with cerebrovascular stroke is the second leading cause of disability and death worldwide [1]. In many cases, residual sensory and motor deficits remain throughout a lifetime, degrading communication, reducing quality of life and safety, and predisposing stroke survivors to numerous health complications [2,3]. A major limiting factor in the recovery of daily activities after cerebral infarct is the impaired regulation of muscle force that is essential for all types of movement [4,5]. Dysfunction during intricate movements of the fingers or orofacial muscle groups is often prominent in post-stroke survivors, as brain lesions that damage cortical upper motor neurons also degrade motor units through trans-synaptic mechanisms, particularly in high density, monosynaptic motoneuron pools which control the fingers of the hand and perioral structures [6,7,8,9,10].

Ischemic brain lesions can adversely affect hand and digits during precision grip or pinch force stability, and movement accuracy during object manipulation and reach [11,12,13]. Following an infarct in the territory of the middle cerebral arteries (MCA) which feed the elaborated sensorimotor cortical representations of the hand and fingers, a loss of higher-order motor planning and sensorimotor integration greatly disrupts fine force production, particularly during individuation of the digits and strength. Secondary degeneration of the corticospinal tract and spinal segment motor unit loss occur in the days and weeks after ischemic injury, also contributing to permanent deficits in kinematic performance [14,15,16,17]. In spared connections, discharge rate coding of recruited motor neurons can be significantly impaired, resulting in abnormal and irregular synaptic input. This, combined with the loss of descending modulatory control, often results in spasticity and contraction instability during attempts at highly controlled fine force production [18,19,20,21].

Similarly, fine motor performance of the lower face and perioral structures are often negatively impacted in stroke and upper motor neuron syndrome, leading to difficulties managing oral intake, reduced speech intelligibility, and impaired facial gesture [10,22,23,24,25]. Perioral and lip force recruitment and end-point control also relies on an elaborated orofacial sensorimotor cortical representation (i.e., Brodmann areas 4,3,1,2) adjacent to the hand-digit cortex and also located in MCA territory. Significant impact on key trigeminofacial feedback systems accompanies motor unit aberrancies, and frequently leads to somatosensory scaling errors during fine orofacial force performance [22,23,26,27]. Difficult to ascertain in standard motor-speech assessment, decrements in perioral force encoding combined with a loss of continuous phasic adjustments during force production often lead to dissolution in motor control at speech-like force magnitudes [25,28,29,30,31].

Thus, a new technology capable of rapid data capture of residual force dynamics and visualization among skilled muscle systems would provide clinicians with enhanced assessment and individualized treatment options for survivors of cerebrovascular stroke in the subacute phase (one week to one month post-infarct). Ideally, this technology should utilize wireless sensing technology to permit bedside transduction of active force dynamics in hand-digital and/or orofacial muscle systems with provisions for real-time acquisition and analysis of salient dependent measures including maximum force to establish the patient’s residual force operating range for contralesional and ipsilesional muscle groups. Ideally, this same biomechanical assessment tool could be used for biofeedback muscle retraining with designed flexibility to individualize treatment according to the patient’s force dynamics profile. 

In the present study, we demonstrate the application of a new wireless force sensing system [32] to assess a cohort of seven chronic stroke survivors’ ability to regulate isometric force during lower lip compression and finger ‘pinch’ in each hand as rapidly and accurately as possible to low-level targets (0.25, 0.5, 1, and 2 N) during a visuomotor ‘ramp-and-hold’ paradigm. This same force sensing system is then used to measure maximum voluntary contraction force (MVCF) levels to establish the force operating range for each muscle group, and finally, contrast these dynamics and MVCF with the same measures obtained from a matched group of neurotypical controls using linear mixed modeling. 

## 2. Materials and Methods

### 2.1. Participants 

Seven chronic (mean = 70 months post-stroke) hemiparetic male MCA stroke survivors (mean = 46.7, SD = 20.1, range = 23–67 years), and 25 neurotypical adult males (mean = 30.0, SD = 14.9, range = 19–65 years). Power analysis was conducted using the average intraclass correlation (0.08) calculated from previous studies—the results indicated that this sample (*N* = 32) would provide adequate power (i.e., ≥80%) for detecting an effect as small as *δ* = 0.26 (small). Written informed consent, approved by the University of Nebraska Institutional Review Board, was obtained for each participant. This study was carried out following the rules of the Declaration of Helsinki of 1975 (https://www.wma.net/what-we-do/medical-ethics/declaration-of-helsinki/), revised in 2013. Stroke participant inclusion criteria were chronic unilateral infarct of MCA territory primarily affecting sensorimotor function, confirmed by medical record and 3T anatomical MRI, 20/20 corrected vision, and ability to follow instructions and perform the biomechanical visuomotor tracking task. Exclusion criteria were total paralysis of contralesional upper limb or face, dementia, and/or aphasia). Gross motor impairment was evaluated in each stroke participant using the Fugl-Meyer Assessment Upper Extremity (FMA-UE) prior to force dynamics assessment. A structural T1-weighted three-dimensional image of the subject’s brain (MPRAGE, Magnetization-Prepared Rapid Gradient-Echo) was acquired for each stroke survivor—TR = 2400 ms, TE = 3.37 ms, voxel size = 1 × 1 × 1 mm, flip angle = 7 degrees, number of slices = 192, acquisition matrix = 256 × 256, field of view (FoV) = 256 × 256 mm, total acquisition time = 5:35 min (clinical features for the stroke survivors are summarized in Table 1).

### 2.2. Instrumentation

A wireless Windows-based (WIN10 ×64) data acquisition and stimulus control system developed in our laboratory (ForceWIN10) was used for visuomotor tracking of participant-generated isometric force signals, as detailed by Greenwood et al. [32]. Wireless strain gage sensors were designed to sample lower lip compression force, and thumb-index finger pinch force using a custom designed Bluetooth low-energy (BLE) data acquisition circuit module which provided bridge excitation, signal conditioning, and analog-to-digital (ADC) conversion (Figure 1, top row). The circuit is a wireless, battery-powered device with a BLE radio, two-stage amplifier, and 24-bit ADC. The first amplification stage can be set by either a potentiometer or resistor to any gain between 5 and 10,000 V/V with a common-mode rejection ratio (CMRR) of at least 83 dB. The ADC has a programmable gain array (1, 8, 64, or 128 V/V) which forms the second stage of amplification. The total amount of amplification is between 5 and 128,000 V/V with 24 bits of resolution. Accuracy is limited primarily by the noise of the first stage amplifier. As a result, a low-noise amplifier was chosen with a noise level of 35 nV/sqrtHz. Conversions are performed 120/s and collected by the Simblee microcontroller unit (MCU) which also contains the BLE radio. 

The firmware was designed to minimize power draw while ensuring data samples are not dropped. When idle, the device broadcasts its pairing information five times per second for quick detection by the personal computer. It consumes 0.78 milliamps (mA) in idle mode giving approximately 500 h of idle time on a 400 mAh battery (3.7 V nom). When connected to a PC via Bluetooth, the device sends 120 samples per second, four samples/packet, with a packet number to ensure correct ordering. During active operation, the device consumes 4.2 mA, and an extra 4.7 mA and 9.4 mA for half- and full-bridge circuits, respectively, given a nominal strain resistance of 350 ohms. The device can operate for 83 h using a half-bridge and 29 h using a full-bridge strain gage circuit.

The thumb-index finger pinch forces were transduced by a Cooper Instruments load cell (Model LKCP 410-25 lb; Warrenton, VA, USA) and conditioned by a separate BLE wireless signal conditioning module worn on the wrist. This load cell has a maximum force of 111.2 N with a 50% overload safe operating range, and a sensitivity of 0.0348 mV/N. The lip cantilever includes integrated fixtures for anterior-posterior and inferior-superior beam translation to accommodate varying occlusal (bite) profiles and maxillofacial anatomy among participants. The lip force cantilever beam is instrumented with a pair of strain gages, one on top and one on bottom, and wired as a two-arm active bridge. The strain sensitivity of the lip cantilever is 50.78 µƐ/N and the overall sensitivity is 0.287 mV/N. The load-sensitive lip cantilever is referenced to a jaw cantilever with scalable 3D-printed titanium jaw dental trays to fit individual maxillofacial anatomy. A dental impression compound (Kerr Extrude XP, Kerr Corporation, Romulus, MI, USA) was used to create a custom mold for the maxillary and mandibular trays. Following a 2-minute curing period, the resulting dental mold provided a comfortable and stable mounting platform for each participant to generate active force on the lower lip force cantilever. Each transducer type was powered by an integrated rechargeable miniature Li-ion (LIB) battery.

To validate the operation of the device, the mean, variance, and linearity were measured (Table 2). First, the instrument was powered for 10 min before a two-point calibration at 0 g = 0 N and 202 g = 2 N was performed. Then, 20 replications of 5-s samples were collected at 0, 20, 50, 100, 200, and 500 g loads on a level surface insulated from vibrations and air drafts. Linearity ranged from 0.9994 to 1.000 and was measured by fitting the means to a line and calculating the R-squared coefficient.

### 2.3. Protocol

This experimental study utilized a factorial design, with one between-subject factor (stroke) and two within-subject factors (muscle group, isometric forces). In a single 30-minute session, participants were oriented to the instrumentation and asked to generate a series of low-level ramp-and-hold isometric forces (0.25, 0.5, 1, and 2 N) in a visuomotor tracking paradigm during lower lip compression, and right- and left thumb-index finger pinch in a randomized block design [33]. Participants were instructed to generate the isometric contractions with the instrumented muscle group ‘as rapidly and accurately as possible’ to the target cursor displayed on a high-definition monitor and sustain the prescribed level of force until a computer-generated auditory cue signaled the end of each 5-s trial. Each structure was tested separately and independently. Ten consecutive trials with randomly varying intertrial intervals (0.5–3 s) were obtained at each of the four target force levels. Participants were encouraged to relax during the intertrial intervals. This was followed by three MVCF trials during which participants were instructed to generate maximum contractions 2 s in duration using the selected muscle system followed by a period of relaxation. Independent variables were target force level, muscle group, and age. Dependent measures included force reaction time (RT), dF/dt_max_ and peak force during recruitment, mean, SD and criterion percentage during hold-phase, and MVCF. Data collection for each sensor-muscle group combination resulted in 43 isometric force trials completed in approximately 7 min. 

### 2.4. Force Signal Processing

All digitized force trials were conditioned by a finite impulse response filter (FIR, low-pass @ 40 Hz) which features high stopband attenuation and a flat passband to retain biomechanically-relevant content. Baseline force was calculated as the mean value of the first 100 ms for each isometric force trial. The hold-phase of each isometric contraction was divided into symmetric analysis epochs designated as T1 and T2. The T1 epoch includes force data samples between 2.0 s and 3.4 s, and the T2 epoch includes data samples between 3.4 s and 4.8 s. The maximum rate of force change (N/s) during recruitment was calculated as the first derivative force maxima (dF/dt_max_). The peak force attained during recruitment from baseline was defined as the maximum force value occurring during the initial 2-s of a ramp-and-hold force trial. Reaction time (ms) is the linearly interpolated value at the point when baseline force is 10 standard deviations (SD) above the baseline. The mean and variance were calculated during each force hold phase epoch. Lastly, hold-phase force criterion percentage is defined as the proportion of the participant’s force signal during the T1 or T2 epochs that are within ±5% of the prescribed target force. 

### 2.5. Statistical Analysis

The thumb-index finger force data for the stroke survivors was transformed to contralesional ‘right-side’ and ipsilesional ‘left-side.’ No transformation was applied to the lower lip force data since this was a midline compression force maneuver. Linear mixed modeling (LMM) was then conducted for each dependent variable to examine differences between participant groups (stroke-affected, neurotypical controls; i.e., group effect), muscle groups (right thumb-index finger, left thumb-index finger, lower lip midline; i.e., muscle effect), and target forces (0.25, 0.5, 1, 2 N; i.e., force effect), as well as muscle differences by participant group (i.e., muscle-by-group interaction effect) and by target force (i.e., muscle-by-force interaction effect). The models accounted for participants’ age and handedness, and dependency of observations repeated within participants (i.e., intraclass correlation). Restricted maximum likelihood (REML) estimation was employed to produce unbiased estimates of the model effects in the unbalanced sample. Statistical significance of the effects was determined at 0.05 alpha level. When the muscle, muscle-by-group, and/or muscle-by-force effect was significant, marginal means were pairwise compared at a Bonferroni-corrected alpha level while controlling for Type I error at the nominal level. A proper error covariance structure was determined for each dependent variable based on model fit (i.e., adjusted Akaike information criterion, Bayesian information criterion). The analysis was performed using SAS 9.4 [34].

## 3. Results

A full series of ramp-and-hold force trials during thumb-index finger pinch (left and right), and lower lip compression are shown in Figure 1 for a 63 year-old neurotypical male (middle panel) and a 66 year-old male (bottom panel) who survived a right MCA ischemic stroke. The dissolution in isometric force performance is apparent as a function of target force (0.25 [blue], 0.5 [red], 1.0 [green], and 2.0 N [magenta]), especially for the contralesional digits of the hand and to a lesser extent in the ipsilesional digits and lower lip. Among the seven stroke participants, a myriad of significant effects were documented for ipsi- and contralesional muscle systems, and between stroke-affected and neurotypical adults using LMM. These are summarized below. 

The contralesional thumb-index finger manifest increased force reaction time compared to other muscle groups and control participants as shown in Figure 2. LMM for reaction time revealed a significant interaction between stroke and muscle group after controlling for participant’s age and handedness (*F*(2,3760) = 27.25, *p* < 0.0001). Post-hoc pairwise comparisons further revealed that the stroke-affected right thumb-index finger (contralesional) manifest a significantly increased RT (marginal mean of 650 ms) compared to stroke-affected left thumb-index finger (530 ms, *p* < 0.0001, d = 0.175) and lower lip (490 ms, *p* < 0.0001, d = 0.162). Additionally, there was a significant interaction between force target and muscle group after controlling for participant’s age and handedness (*F*(6,3760) = 3.74, *p* < 0.01). Among the four target forces, there was a tendency for the right thumb-index finger to show increased variance and elevated mean and median values compared to the left thumb-index finger for the dependent variable ‘force reaction time’ with the longest RTs observed at the 0.25 N target. Post-hoc pairwise comparisons revealed this pattern between the right and left digit pinch force RTs at FT 0.25 N (right 620 ms versus left 560 ms, *p* < 0.01, d = 0.094) and FT 1 N (right 610 ms versus left 530 ms, *p* < 0.0001, d = 0.116). 

Stroke patients showed a tendency to overshoot peak isometric force among the three muscle groups during recruitment (Figure 3), with the contralesional digits showing greater overshoot during pinch compared to ipsilesional digits and the lower lip compression. The LMM analysis revealed a significant interaction between stroke and muscle group after controlling for participant’s age and handedness (*F*(2,3796) = 7.60, *p* < 0.001). Post-hoc pairwise comparisons further revealed that ipsi- and contralesional thumb-index finger pinch peak forces were significantly higher (estimated marginal means of 1.82 and 1.85 N, respectively) compared to neurotypical controls (estimated marginal means of 1.51 and 1.49 N, respectively) for these muscle systems (adjusted *p* < 0.05, d = 0.094; adjusted *p* < 0.05, d = 0.103). Additionally, there was a significant interaction between force target and muscle group after controlling for participant’s age and handedness (*F*(6,3796) = 4.32, *p* < 0.001). As expected, peak isometric force increased with increases in target force. Post-hoc pairwise comparisons further revealed that the FT 2 x left thumb-index finger combination showed the highest peak force (estimated mean = 3.09 N), which was significantly higher than the peak force in other force target x muscle group combinations (all adjusted *p* < 0.05, d = 0.108–1.024). 

Stroke patients showed some differences among muscle groups for peak rates of force recruitment (dF/dt_max_) for stroke survivors (Figure 4). The LMM analysis for dF/dt_max_ revealed a significant interaction between stroke and muscle group after controlling for participant’s age and handedness (*F*(2,3796) = 8.75, *p* < 0.001). Post-hoc pairwise comparisons further revealed that the stroke-affected left thumb-index finger group (ipsilesional) resulted in the highest dF/dt_max_ at 26.99 N/s, which was significantly greater than the maximum rate of force recruitment profiles in some of the other stroke and muscle group combinations, including the stroke-affected lower lip (adjusted *p* < 0.0001, d = 0.173), and stroke-affected right thumb-index finger (adjusted *p* < 0.001, d = 0.142). Additionally, there was a significant interaction between force target and muscle group after controlling for participant’s age and handedness (*F*(6,3796) = 2.41, *p* < 0.05). Post-hoc pairwise comparisons further revealed that the FT 2 target yielded the highest dF/dt_max_ values among muscle groups (left-hand digits = 35.63 N/s, right-hand digits = 31.22 N/s, lower lip = 29.49 N/s). The dF/dt_max_ values decreased as force target decreased from 2 N to 0.25 N. Compared to the 2 N performance, dF/dt_max_ values among muscle groups at FT 0.25 N decreased by more than 60% (left-hand digits = 17.33 N/s, right-hand digits = 15.33 N/s, lower lip = 15.90 N/s). LMM confirmed the trend of dF/dt_max_ declination among muscle groups as a function of target force—all adjusted *p* < 0.0001, *d* = 0.209–0.440 for left thumb-index finger group; with similar trends for the right thumb-index finger group, all adjusted *p* < 0.0001, d = 0.189-0.382; and all adjusted *p* < 0.0001, d = 0.239–0.326 for the lower lip. At the FT 2 N target level, the left thumb-index finger manifest significantly higher dF/dt_max_ values compared to the lower lip (adjusted *p* < 0.001, d = 148).

Stroke patients showed slightly elevated mean force compared to neurotypical adults for the digits during the T1 hold phase, whereas the lower lip manifested a small undershoot to target for both stroke and neurotypical participants. This error in digit force output was diminished during the T2 hold phase (Figure 5). For T1 hold phase mean force, LMM revealed a significant interaction between stroke and muscle group after controlling for participant’s age and handedness (*F*(2,3796) = 6.06, *p* < 0.01). Post-hoc pairwise comparisons further revealed that the stroke-affected (contralesional and ipsilesional) thumb-index fingers yielded a significantly higher T1 hold phase mean force output compared to the stroke-affected lower lip (adjusted *p* < 0.0001, d = 0.158; adjusted *p* < 0.0001, d = 0.157, respectively). Finally, the neurotypical thumb-index fingers manifest significantly higher T1 hold phase forces compared to their lip muscle group (all adjusted *p* < 0.0001, d = 0.034–0.046). Additionally, there was a significant interaction between force target and muscle group after controlling for participant’s age and handedness (*F*(6,3796) = 7.50, *p* < 0.0001). Post-hoc pairwise comparisons further revealed that the stroke-affected (contralesional) thumb-index finger yielded a significantly higher T1 hold phase mean force output at FT 2 N compared to all other FT and muscle combinations (all adjusted *p* < 0.0001, d = 0.126–3.137).

For T2 hold phase mean force, LMM revealed a marginally significant interaction between stroke and muscle group after controlling for participant’s age and handedness (*F*(2,3796) = 2.92, *p* = 0.054). Post-hoc pairwise comparisons further revealed that the stroke-affected (contralesional and ipsilesional) thumb-index fingers yielded a significantly higher T2 hold phase mean force output compared to the stroke-affected lower lip (adjusted *p* < 0.0001, d = 0.158; adjusted *p* < 0.0001, d = 0.133, respectively). The stroke-affected fingers (contralesional and ipsilesional) also produced higher T2 hold phase mean force compared to the neurotypical lip (adjusted *p* < 0.01, d = 0.113; adjusted *p* < 0.05, d = 0.097, respectively). Finally, the neurotypical thumb-index fingers manifest significantly higher T2 hold phase forces compared to their lip muscle group (all adjusted *p* < 0.0001, d = 0.039–0.051). Additionally, there was a significant interaction between force target and muscle group after controlling for participant’s age and handedness (*F*(6,3796) = 7.29, *p* < 0.0001). Post-hoc pairwise comparisons further revealed that the stroke-affected (contralesional) thumb-index finger yielded a significantly higher T2 hold phase mean force output compared to all other FT and muscle combinations (all adjusted *p* < 0.0001, d = 0.136–3.139).

Stroke patients showed substantial increases in force standard deviation (SD) relative to the specified force target, especially for the digits of the contralesional hand followed by the ipsilesional hand, with smaller changes observed in the lower lip the hold-phase periods T1 and T2 compared to adult controls (Figure 6). For T1 hold phase standard deviation (SD) [force data samples between 2.0 s and 3.4 s], there was a significant interaction between stroke and muscle group after controlling for participant’s age and handedness (*F*(2,3796) = 54.55, *p* < 0.0001). Post-hoc pairwise comparisons further revealed that the estimated marginal means for the contralesional (right thumb-index finger) digits of stroke-affected adults showed the highest SD (0.24 N) among all participant and muscle group combinations, significantly greater than neurotypical muscle groups including the right thumb-index finger (*p* < 0.0001, d = 0.197), left thumb-index finger (*p* < 0.0001, d = 0.192), and lower lip (*p* < 0.05, d = 0.099). The contralesional thumb-index finger SD was marginally greater than ipsilesional right thumb-index finger (0.24 N vs. 0.20 N, *p* = 0.0547, d = 0.092). Overall, the thumb-index finger digits among stroke participants manifest greater instability in regulating fine force compared to their neurotypical counterparts [right side 0.24 vs. 0.12 N, left side 0.20 vs. 0.12 N] (ipsilesional, adjusted *p* < 0.001, d = 0.136; contralesional, adjusted *p* < 0.0001, d = 0.197). Additionally, there was a significant interaction between force target and muscle group after controlling for participant’s age and handedness (*F*(6,3796) = 2.57, *p* < 0.05). The absolute contractile instability tends to increase with increasing target force levels. The estimated marginal means are slightly higher for the lower lip (0.26 N) compared to either thumb-index finger combination (right 0.23 N, left 0.24 N) at FT 2. Pairwise comparisons further showed that at 2 N, the lower lip resulted in significantly higher T1 hold force SDs which was significantly greater than hold force SDs in all other lesser FT by muscle combinations (all adjusted *p* < 0.0001, d = 0.217–0.368).

For T2 hold phase SD (force data samples between 3.4 s and 4.8 s), there was a significant interaction between stroke and muscle group after controlling for participant’s age and handedness (*F*(2,3796) = 60.41, *p* < 0.0001). Post-hoc pairwise comparisons further revealed that the estimated marginal means for the contralesional (right thumb-index finger) digits of the hand among stroke-affected adults showed the highest SD (0.17 N) among all participant and muscle group combinations. The stroke-affected left thumb-index finger manifest significantly greater hold force SD when compared to the stroke-affected lower lip (0.15 N vs. 0.10 N, adjusted *p* < 0.0001, d = 0.135). Overall, the thumb-index finger digits among stroke participants manifest greater instability in regulating fine force compared to their neurotypical counterparts [right side 0.17 vs. 0.07 N, left side 0.15 vs. 0.07 N] (contralesional, adjusted *p* < 0.0001, d = 0.172; ipsilesional, adjusted *p* < 0.0001, d = 0.140). No significant difference was found between stroke-affected and neurotypical lip force SDs. Additionally, there was a significant interaction between force target and muscle group after controlling for participant’s age and handedness (*F*(6,3796) = 6.26, *p* < 0.0001). The absolute level of contractile instability tends to increase with increasing target force levels during the T2 hold phase. The estimated marginal means for SD were slightly higher for the lower lip (0.18 N) compared to either thumb-index finger combination (right 0.15 N, left 0.17 N) at FT 2. Pairwise comparisons further showed that at 2 N, the lower lip resulted in significantly higher hold force SDs at FT 2 compared to hold force SDs in all other FT by muscle combinations (all adjusted *p* < 0.0001, d = 0.099–0.335), except for the FT 2 by left thumb-index finger combination.

Stroke patients showed a fundamental impairment in the ability to maintain isometric force levels within a prescribed target window at ±5%, regardless of muscle group. The most striking degradation in hold-phase criterion performance was observed in pinch force control by the contralesional thumb-index finger followed by the ipsilesional hand and lower lip during the hold-phase periods T1 and T2 compared to adult controls (Figure 7). For T1 hold phase force criterion percentage (force data samples between 2.0 s and 3.4 s), LMM analyses revealed a significant interaction after controlling for age and handedness between participant groups and muscles (*F*(2,3796) = 47.49, *p* < 0.0001). Overall, the ability to regulate force within a ±5% criterion window for each of the force targets was degraded in MCA stroke survivors compared to neurotypical controls. Post-hoc pairwise comparisons further revealed that the contralesional right thumb-index finger of stroke-affected adults showed the lowest hold force criterion percentage (only 20.84% of force output was within ±5% of endpoint target). Moreover, the stroke-affected right thumb-index finger hold phase force performance was worse than hold force performance in the stroke-affected left thumb-index finger (20.84 vs. 35.63%, adjusted *p* < 0.0001, d = 0.161), and both the right and left thumb-index fingers in neurotypical adults (right 45.70%, adjusted *p* < 0.0001, d = 0.165; left 46.08%, adjusted *p* < 0.0001, d = 0.167). Additionally, there was a significant interaction after controlling for age and handedness between force targets and muscles (*F*(6,3796) = 10.64, *p* < 0.0001). In general, criterion performance improved with increasing target force levels, ultimately attaining 32.59% (lower lip), 55.25% (left thumb-index finger), and 52.84% (right thumb-index finger) at the 2 N target force. The performance advantage for the fingers over the lower lip was apparent at FT 0.5, FT 1, and FT 2 target levels, except at FT 0.25 where criterion performance was nearly equivalent for the right thumb-index finger (12.98%) and lip (12.53%). At the FT 0.25 target force, the left thumb-index finger manifest higher criterion performance compared to the right thumb-index finger (21.95%, adjusted *p* < 0.01, d = 0.100) and lower lip (12.53%, adjusted *p* < 0.001, d = 0.104).

For T2 hold phase force criterion percentage (force data samples between 3.4 s and 4.8 s), there was a significant interaction after controlling for age and handedness between participant groups and muscles (*F*(2,3796) = 39.53, *p* < 0.0001). Overall, the ability to regulate force within a ±5% criterion window for each of the force targets was degraded in MCA stroke survivors compared to neurotypical controls. Post-hoc pairwise comparisons further revealed that the right thumb-index finger of stroke-affected adults showed the poorest hold force criterion percentage (24.37%) compared to the same muscle system in neurotypical controls (50.44%; adjusted *p* < 0.0001, d = 0.156). T2 hold phase criterion performance for the left thumb-index finger of stroke-affected adults (40.79%) again was lower compared neurotypical (53.64%) but did not reach statistical significance. No significant difference between participant groups was found for the lower lip (stroke-affected 28.72% vs. neurotypical 32.46%). Additionally, there was a significant interaction after controlling for age and handedness between force target and muscles (*F*(6,3796) = 13.25, *p* < 0.0001). Similar to the T1 hold phase findings, criterion performance during the T2 hold phase improved as a function of target force, ultimately attaining 39.17% (lower lip), 65.10% (left thumb-index finger), and 58.87% (right thumb-index finger) at the 2 N target force. Compared to T1 hold phase measurements, this represents an increase of 20.19%, 17.83%, and 11.41%, respectively. Post-hoc pairwise comparisons further revealed that the FT 2 × left thumb-index finger combination showed the highest criterion force percentage (estimated mean = 65.10%), which was significantly higher than the criterion percentage at the lower force targets (1, 0.5, and 0.25 N) × muscle group combinations (all adjusted *p* < 0.01, d = 0.089–0.407). The later occurring T2 hold phase epoch affords the participant additional time to adjust force regulation to achieve higher levels of ‘on target’ performance in each muscle system. The performance advantage for the right and left digits over the lower lip was apparent at FT 0.5 N (30.75%, 43.87%, and 29.53%, respectively), FT 1 N (45.87%, 56.26%, and 36.32%, respectively), and FT 2 N (58.87%, 65.10%, and 39.17%, respectively) target levels, except at FT 0.25 where criterion performance for the lip (17.34%) and right thumb-index finger (14.13%) were statistically equivalent.

Accompanying the myriad of impairments in fine force dynamics previously described, we found MVCF levels greatly diminished for the digits during pinch, and spared for lower lip compression (Figure 8). The LMM analyses revealed a significant weakness during maximum pinch force production that was greatest in the contralesional hand (right side), whereas no significant difference was found for lower lip MVCF between stroke and control participants (14.58 vs. 11.11 N, respectively). For MVCF there was a significant interaction between stroke and muscle group after controlling for participant’s age and handedness (*F*(2,252) = 46.63, *p* < 0.0001). Post-hoc pairwise comparisons further revealed that the stroke-affected right thumb-index finger (contralesional) yielded significantly lower MVCF compared to control right thumb-index finger with a moderate effect size (15.93 vs. 45.82 N, adjusted *p* < 0.0001, d = 0.628). The contralesional thumb-index finger also showed greater loss in MVCF compared to the ipsilesional thumb-index finger (15.93 vs. 27.23 N, adjusted *p* < 0.01, d = 0.380).

## 4. Discussion

The elaborate repertoire of movements displayed by distal limb and orofacial structures is dependent on adaptive regulation of muscle force recruitment and postural motor control [33,35,36,37]. Precision grip and object manipulation, speech, deglutition, facial expression, all rely on highly regulated force dynamics which occur in the presence of sensorimotor mechanisms to rapidly adapt to external loads [38,39,40,41,42].

Degradation of upper extremity [43,44] and orofacial function [24,25,45,46] is common among survivors of cerebrovascular stroke. The sequelae associated with stroke affecting the MCA territory often results in a myriad of impairments that disproportionately affect distal limb force and kinetics (hand-digits) [44,47,48], and orofacial movements [24], along with diminished somatosensory detection thresholds [46] and cognitive function [45]. For example, impairments in force generation and regulation are associated with a reduction in corticomotoneuron output, disordered extrapyramidal subcortical control mechanisms, altered reflex mechanisms, and muscle atrophy [25,49]. Collectively, these sensorimotor performance limitations significantly affect the quality of life [50,51,52].

The ramp-and-hold task dynamic has been applied in the present study among our cohort of MCA stroke and neurotypical participants to assess their ability to recruit isometric force using the lower lip, right thumb-index finger, and left thumb-index finger as “rapidly and accurately as possible” to low-level targets ranging from 0.25 to 2 N using wireless force sensors. The selected range of low-level forces is comparable to ‘skilled’ force levels used previously for study of thumb-index finger pinch regulation [33,53,54] and perioral biomechanics [27,33,39,42,55,56]. At these force levels, mechanoreceptor activity and reflex sensitivity is robust during precision two-finger grip [54,57,58], and cyclic lip movements typical of bilabial syllable production and lip contact [31,59,60,61].

The application of our custom wireless force sensors [32] to rapidly assess fine force dynamics in adult survivors of MCA stroke revealed dissolution in both force regulation during ramp-and-hold dynamics and maximum voluntary force generation. Overall, the contralesional thumb-index finger muscle group manifested more significant force control impairments compared to the ipsilesional digit muscle group. The stroke-affected muscle groups showed degraded force performance relative to neurotypical controls on a number of dependent variables, including (1) force reaction time (>110 ms increase in RT for contralesional digit compared to ipsilesional digit and lower lip); (2) observed a 34.4% increase in peak force relative to controls accompanied by an increase in dF/dt_max_; (3) elevated mean force during T1 and T2 hold phase periods, approximately 200% to 350% increase in hold phase SD relative to controls during T1 and T2 hold phase periods; and (4) marginal end-point control during the T1 and T2 hold phases with only 24% of their force waveform data points during visuomotor tracking within the ±5% target force criterion window, compared to ~50% criterion performance among neurotypical controls. Smaller departures from normal fine force regulation were associated with the lower lip midline compression maneuver.

Peak force overshoot has been observed in neurotypical adults performing object prehension using the thumb-index finger [33,62] and during lower lip isometric compression maneuvers [26,33]. This phenomena may reflect a motor control strategy to produce an acceptable tradeoff between the rate of force recruitment (dF/dt_max_) and force end-point accuracy. The exaggerated error in peak force control among the present cohort of MCA stroke survivors likely reflects an inability to precisely balance the agonist drive required during force recruitment and decruitment to the hold phase. Similar errors in peak force control have been reported in patients who have sustained traumatic brain injury [26,63], Parkinson’s disease [64], and amyotrophic lateral sclerosis [65,66].

The MVCF task revealed a significant weakness in the digits, especially in the contralesional hand (thumb-index finger) which manifest a 74% reduction [12.15 N vs. 44.86 N] in pinch force relative to their neurotypical controls. Residual MVCF ipsilesional pinch levels were better preserved than the contralesional, but nonetheless, manifest a modest 40% reduction [23.45 N vs. 39.20 N] in maximum force output compared to their neurotypical controls. Lower lip compression MVCF was equivocal between stroke and neurotypical groups (10.79 N vs. 9.60 N), with substantially lower capacity compared to the hand-digits. This is consistent with anatomical differences in musculoskeletal and mechanosensory anatomy (muscle size, motor unit composition, degrees of freedom, presence/absence of bones and joints, mechanoreceptor typing and density, brain representation, and high cortical magnification factor) between the lip and fingers [33,67,68,69,70].

In the present sample of adult male MCA stroke participants, it is apparent that clinically-reported unilateral MCA infarcts can lead to bilateral impairments in MVCF output and fine force dynamics that are most prevalent in contralesional muscle groups (contralesional > ipsilesional). These findings reinforce the need for bilateral measurement of force dynamics for muscle groups of interest in cerebrovascular stroke.

Bedside evaluation of isometric force dynamics with our portable, laptop-based, wireless data acquisition and stimulus control system can be completed in 7 minutes or less for any ‘skill’ muscle group of interest, and shows promise to chart the topography of force dysregulation in stroke survivors. This sensor system can also be used to map the efficacy of treatment strategies in repeated measure designs, or configured to provide stroke patients with therapeutic visual biofeedback of their isometric force trials in real time. For stroke patients in the subacute phase (one week to one month post-infarction), the use of non-invasive, real-time biomechanical measures as described in the present report provides the clinician with objective measures of residual motor function at low force levels among contralesional and ipsilesional muscle groups which are essential for skilled motor activities of daily living including digital manipulation, speech, and eating. The large dynamic range of our wireless force sensors allows the clinician to measure both skilled, low-level force dynamics as well as maximum voluntary force output in muscle groups of the face and hand. Inclusion of the MVCF measurement yields important clinical data to establish the residual force operating range for each muscle group assessed. Force dynamics assessment also has merit to measure longitudinal change in stroke status over time, or to assess the effectiveness of rehabilitation strategies designed to improve fine motor control (i.e., repetitive somatosensory stimulation of the affected hand, limb, or orofacial structure). The same instrumentation used for force dynamics assessment can be used as a biofeedback tool where criterion-level force targets can be specified by the clinician and individualized for each stroke survivor during motor retraining beginning as early as the subacute phase. Moreover, the sensitivity of biomechanical measures of force dynamics during purposive motor control will serve future studies designed to quantify the efficacy of neuroprotection strategies in hyperacute stroke intervention where one primary goal is to mitigate the evolution of infarct development following ischemia of the MCA territory and its sensorimotor representation in the forebrain [71].

One area of special significance concerns the neural encoding of force within motor cortex and subcortical structures for control of the extremities. The firing patterns of neurons in primary motor cortex (M1) are highly correlated with select parameters of force, especially the rate of force change and absolute static force levels, compared to other movement parameters [72,73,74,75,76]. In non-human primates, a large proportion of pyramidal tract neurons recorded from M1 manifest a stronger relation to the rate of force change during ramp-and-hold isometric contractions than to static force [75]. Additionally, the activity level of corticomotoneurons is increased when the force levels and increments in force output are relatively small and within the force operating range for skilled motor behavior [76,77].

A cerebral network consisting of parietal and frontal regions (including dorsal and ventral premotor cortex, parietal cortex, sensorimotor cortex, and cingulate cortex) has been revealed during grasp and precision grip tasks in humans using PET and fMRI [78,79,80,81,82,83,84,85]. Blood oxygen level dependent (fMRI BOLD) modulation has shown precise force scaling and task specificity during static grip in a cerebral network, including sensorimotor cortex (M1/S1), ventral premotor cortex (PMv), inferior parietal cortex, and supplementary and cingulate motor areas (SMA/CMA) [81]. Controlling force instabilities during manipulation requires specific cortical-striatal-cerebellar networks [49]. A critical role for the cerebellum has been identified in a force field task in which humans have shown the ability to correct for errors during movement, the ability to learn from such errors, and accommodate the resulting motor memory during reach [86]. The present study involving MCA stroke survivors revealed large errors in regulating stable force output to the four endpoint target force levels. It is likely that cerebral infarction involving the MCA territory degraded the integrity of corticostriatal pathways and portions of the cortico-ponto-cerebello-dentato-thalamo-cortical feedback network.

The ramp-and-hold task dynamic has been applied in the present study among our cohort of MCA stroke and neurotypical participants to assess their ability to recruit isometric force using the lower lip, right thumb-index finger, and left thumb-index finger as “rapidly and accurately as possible” to low-level targets ranging from 0.25 to 2N using wireless force sensors. The selected range of low-level forces is comparable to force levels used previously for study of thumb-index finger pinch regulation [33,53,54] and perioral speech biomechanics [27,39,42,55,56]. At these force levels, mechanoreceptor activity and reflex sensitivity is robust during precision two-finger grip [54,57,58], and cyclic lip movements typical of bilabial syllable production and lip contact [31,59,60,61].

Our neurotypical participants were able to regulate thumb-finger pinch forces within the ±5% criterion force window at a significantly higher percentage compared to the lower lip. Conversely, the MCA stroke participants manifest a disproportionately large impairment in end-point and hold-phase criterion force level performance during pinch force generation. The precision of thumb-index finger opposition during pinch force control and manipulation is well documented in normal adults [53]. Additional factors potentially contributing to the observed thumb-index finger advantage include an elaborated representation of muscle spindle afferents and mechanoreceptors for encoding the consequences of force generation and movement. The digits of the glabrous hand are well endowed with Ruffini, Meissner, and Pacinian (PC) corpuscles, and Merkel cells [87]. In contrast, the lower face lacks PC mechanoreceptors [88,89,90] resulting in significant differences in somatosensory function (vibrotactile detection, two-point discrimination) between the glabrous hand [91,92,93] and lower face [59,92,93,94,95,96]. Moreover, lower lip anatomy differs substantially from the digits of the hand. Perioral lip tissues are essential askeletal and consist of a mass of interdigitating muscle fibers, with insertions directly into perioral glabrous and/or hairy skin, or converge into semi-tendinous nodes in the cheek. The lower lip also is kinematically referenced to the moveable mandible [42,55,97]. Proprioceptive afferents (muscle spindles, Golgi tendon organs) which are common to muscles and tendons of the hand and forearm, are absent in muscles of the lower face [98,99]. The encoding of proprioceptive-like cues are thought to originate from specialized Ruffini mechanoreceptor nerve endings in the lips [100].

There is considerable value in the application of biomechanics to study movement disorders. For example, knowledge of force dynamics is central to advance our understanding of multi-joint coordination in the hand in health and disease. Recently, a novel robotic approach has been reported to restore mechanics of functional hand tasks (fingertip pinch force) in stroke survivors by providing an individualized pattern of targeted assistance to impaired muscle groups using non-restraining biomimetic actuators [44]. This approach emphasized the importance of restoring task mechanics, error-based learning, and patient participation. Another interesting approach considers the cortical regulation of agonist and antagonist muscle activations by quantifying the magnitude of corticomuscular coherence which may be relevant in specific populations (stroke, cerebral palsy, cervical spinal cord injury) who manifest impaired antagonist muscle control [101]. The inclusion of biomechanical measures in a clinical setting using portable wireless force sensing technology and automated signal processing methods is expected to promote the development of individualized treatment programs for rehabilitation of orofacial and hand-digit motor skills which are often impaired in cerebrovascular stroke and brain injury [2,11,32,33,62].

Limitations of the present study include the relatively small sample of chronic MCA stroke patients, comparison of stroke patients and neurotypical adults in largely unbalanced sizes, and lower lip force transduction. Plans are underway for a large-scale trial that will include force dynamics and MVCF measures involving 200 hyperacute MCA ischemic stroke patients. We have since developed a new lip force transducer that has been tested in 40 neurotypical adults to allow independent assessment of the right and left oral angle during medial compression ‘lip rounding’ isometric force maneuvers [102]. This will provide a new form of perioral force data to complement the bilateral thumb-index finger pinch force measures.

## Figures and Tables

**Figure 1 sensors-20-01221-f001:**
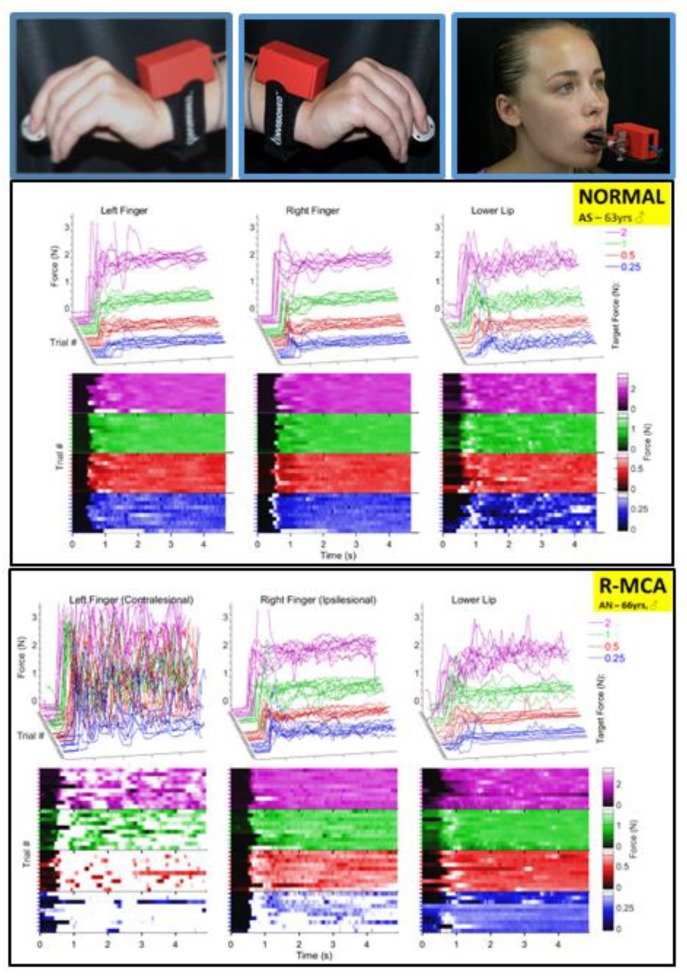
Wireless strain gage sensors (upper row). Left thumb-index finger ‘pinch,’ right thumb-index finger pinch, and lower lip midline ‘compression’ in situ. Middle plot panel shows individual ramp-and-hold force trials at 0.25, 0.5, 1, and 2 N target force levels in a waterfall display format for the left thumb-index finger, right thumb-index finger, and lower lip sampled from a neurotypical 63 year-old male. Bottom plot panel shows the same data types for a 66 year-old male R-MCA (right- middle cerebral artery) stroke survivor. Heat maps are shown below each muscle group time series to illustrate the variability between participants in force onset and absolute force amplitude coded by a color scale.

**Figure 2 sensors-20-01221-f002:**
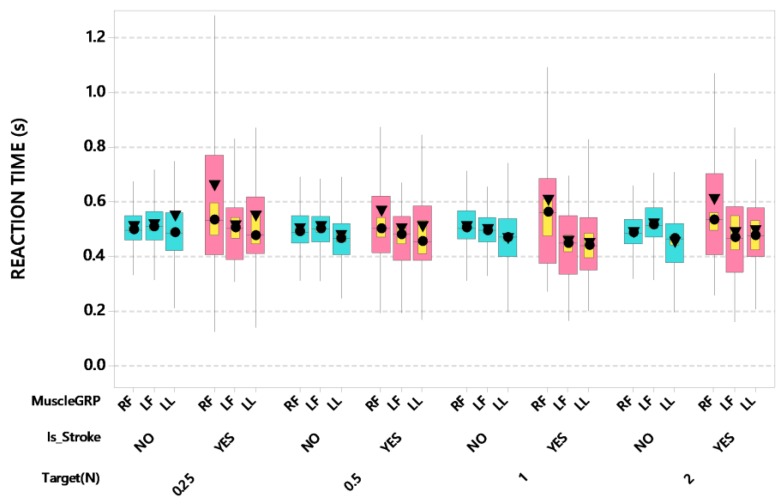
Boxplot of force reaction time (s) as a function of muscle group (RF = right finger [contralesional for stroke participants], LF = left finger [ipsilesional for stroke participants], and LL = lower lip), stroke status, and target force. Interquartile boxes (cyan, pink), 95% confidence interval for the median (yellow), ▼ = mean, ● = median. The whiskers represent the ranges for the bottom 25% and the top 25% of the data values, excluding outliers.

**Figure 3 sensors-20-01221-f003:**
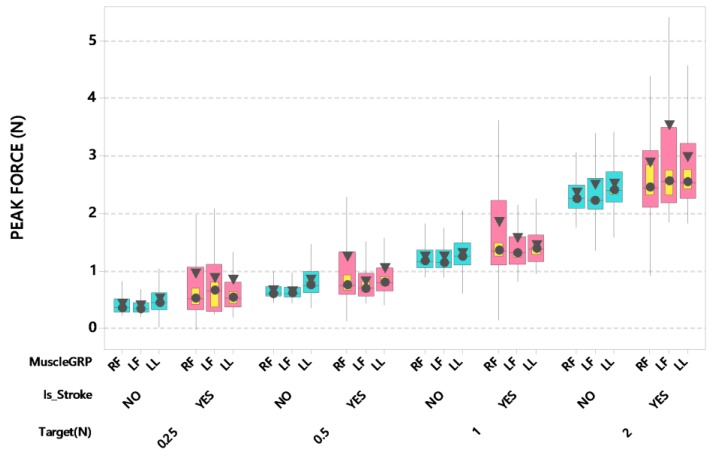
Boxplot of peak force (N) during recruitment as a function of muscle group (RF = right finger [contralesional for stroke participants], LF = left finger [ipsilesional for stroke participants], and LL = lower lip), stroke status, and target force. Interquartile boxes (cyan, pink), 95% confidence interval for the median (yellow), ▼ = mean, ● = median. The whiskers represent the ranges for the bottom 25% and the top 25% of the data values, excluding outliers.

**Figure 4 sensors-20-01221-f004:**
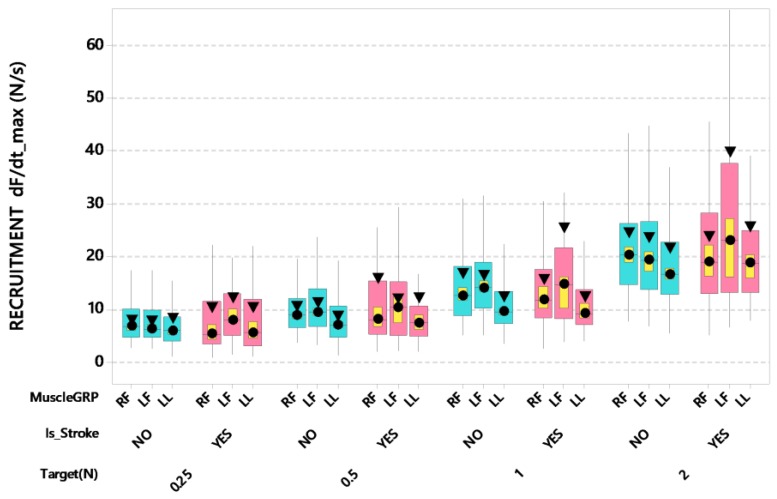
Boxplot of dF/dt_max_ (N/s) during recruitment as a function of muscle group (RF = right finger [contralesional for stroke participants], LF = left finger [ipsilesional for stroke participants], and LL = lower lip), stroke status, and target force. Interquartile boxes (cyan, pink), 95% confidence interval for the median (yellow), ▼ = mean, ● = median. The whiskers represent the ranges for the bottom 25% and the top 25% of the data values, excluding outliers.

**Figure 5 sensors-20-01221-f005:**
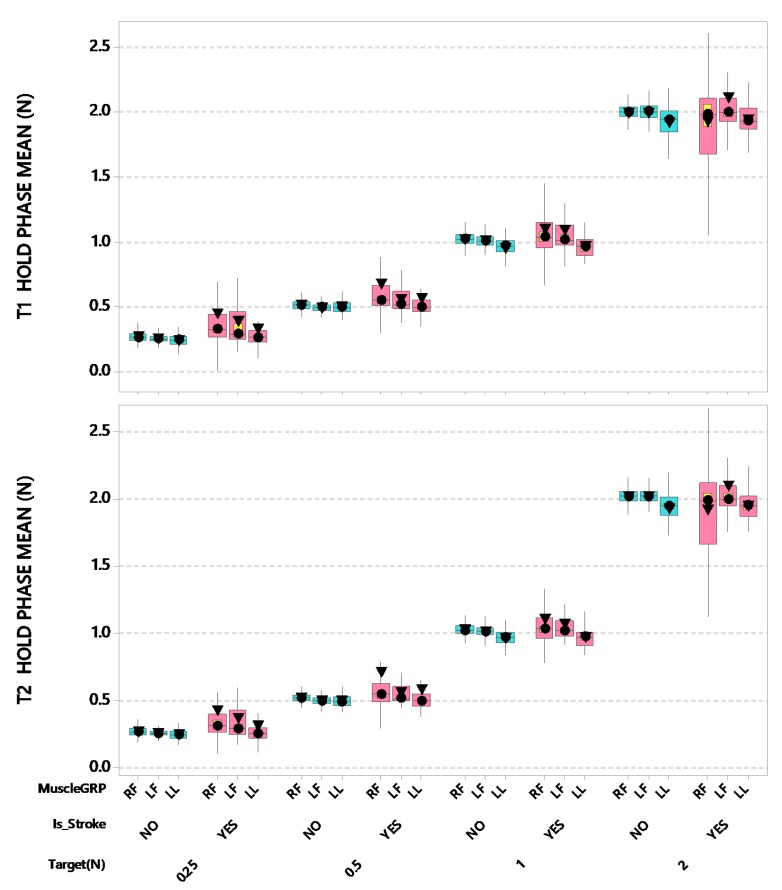
Boxplots of T1 and T2 Hold Phase Mean force (N) as a function of muscle group (RF = right finger [contralesional for stroke participants], LF = left finger [ipsilesional for stroke participants], and LL = lower lip), stroke status, and target force. Interquartile boxes (cyan, pink), 95% confidence interval for the median (yellow), ▼ = mean, ● = median. The whiskers represent the ranges for the bottom 25% and the top 25% of the data values, excluding outliers.

**Figure 6 sensors-20-01221-f006:**
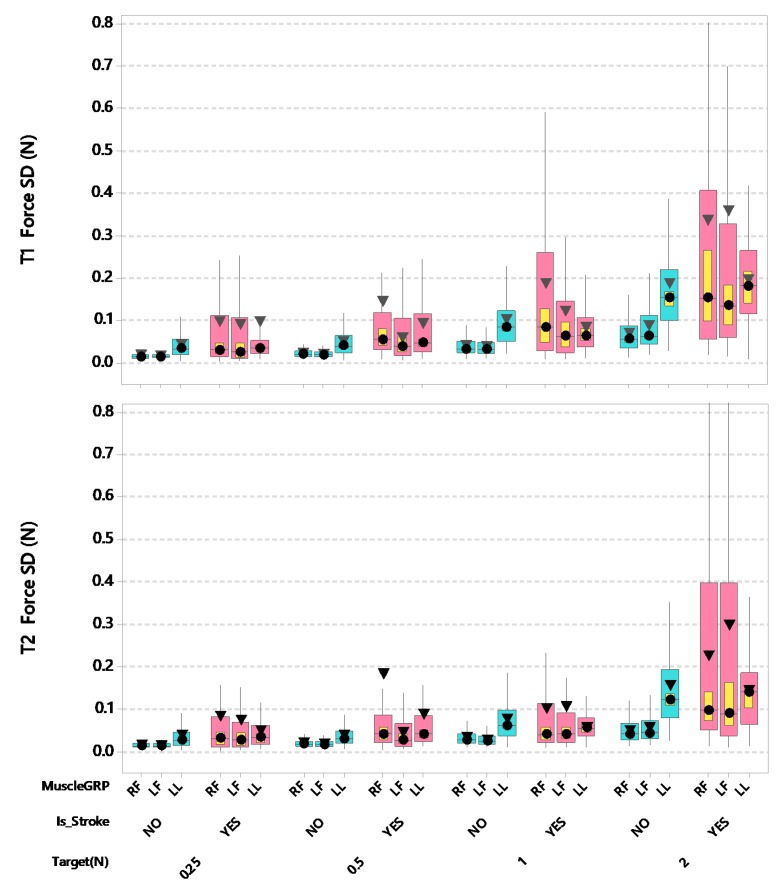
Boxplots of T1 and T2 Hold Phase Force Standard Deviation (N) as a function of muscle group (RF = right finger [contralesional for stroke participants], LF = left finger [ipsilesional for stroke participants], and LL = lower lip), stroke status, and target force. Interquartile boxes (cyan, pink), 95% confidence interval for the median (yellow), ▼ = mean, ● = median. The whiskers represent the ranges for the bottom 25% and the top 25% of the data values, excluding outliers.

**Figure 7 sensors-20-01221-f007:**
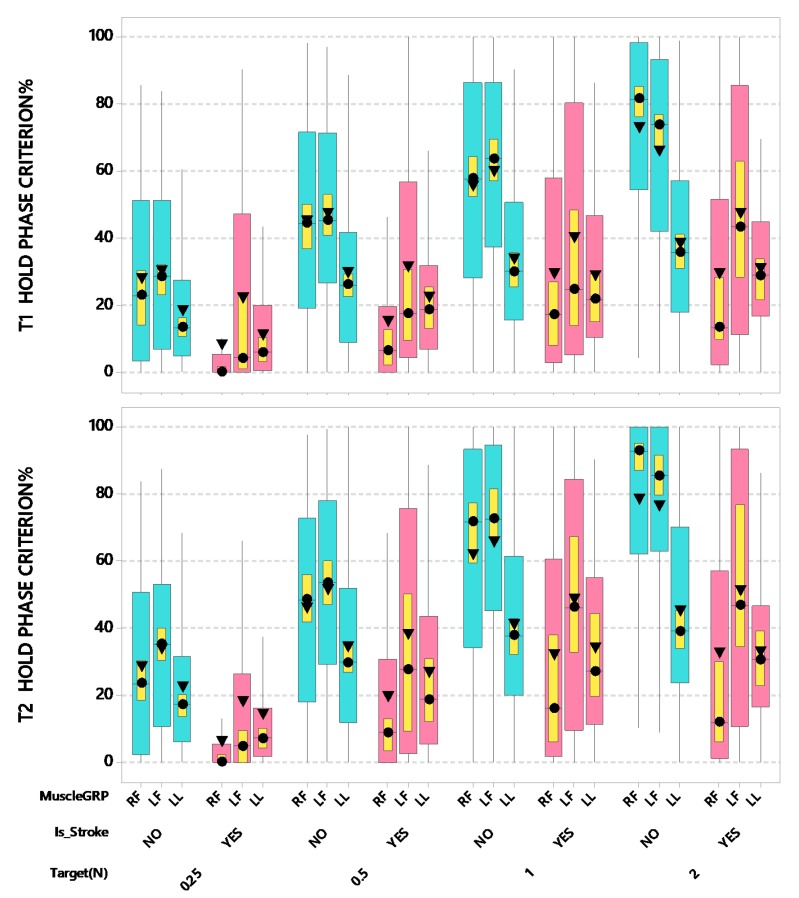
Boxplots of T1 and T2 Hold Phase Criterion Percentage (within a ±5% target force criterion window) as a function of muscle group (RF = right finger [contralesional for stroke participants], LF = left finger [ipsilesional for stroke participants], and LL = lower lip), stroke status, and target force. Interquartile boxes (cyan, pink), 95% confidence interval for the median (yellow), ▼ = mean, ● = median. The whiskers represent the ranges for the bottom 25% and the top 25% of the data values, excluding outliers.

**Figure 8 sensors-20-01221-f008:**
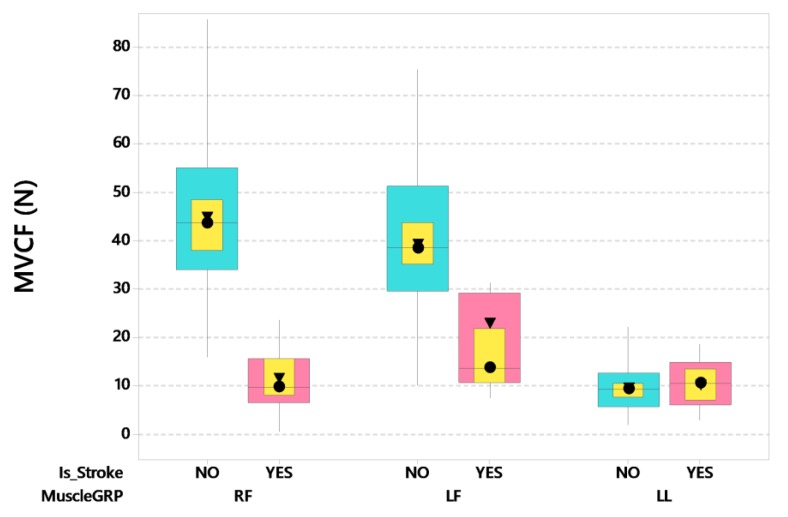
Boxplot of maximum voluntary contraction force (MVCF) as a function of stroke status and muscle group (RF = right finger [contralesional for stroke participants], LF = left finger [ipsilesional for stroke participants], and LL = lower lip). Interquartile boxes (cyan, pink), 95% confidence interval for the median (yellow), ▼ = mean, ● = median. The whiskers represent the ranges for the bottom 25% and the top 25% of the data values, excluding outliers.

**Table 1 sensors-20-01221-t001:** Clinical and brain imaging profile for the seven adult stroke survivors.

ID	Age (yrs)	Sex	Lesion Territory	Post-Stroke (mos)	Anatomic MRI (MPRAGE)			Fugl-Meyer Upper Ext Motor Score (66 Max)	
					Sagittal	Coronal	Transverse	Right	Left
1	67	M	L-MCA	70	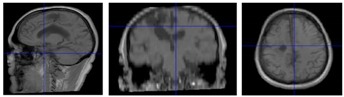			59	66
2	31	M	L-MCA	37	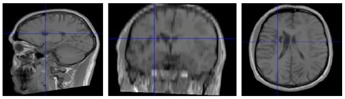			60	66
3	23	M	L-MCA	143	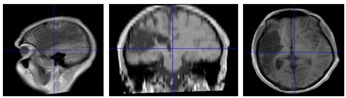			34	65
4	47	M	L-MCA	38	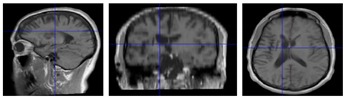			38	66
5	66	M	R-MCA	87	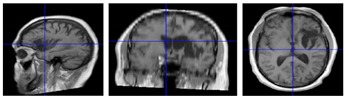			66	24
6	26	M	R-MCA	30	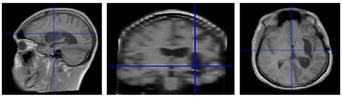			60	33
7	67	M	L-MCA	86	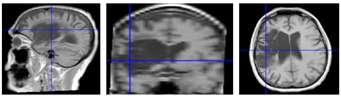			66	66

**Table 2 sensors-20-01221-t002:** Finger and lip strain gage sensor linearity.

	Mass(g)	0	20	50	100	200	500	R^2^
Mean	Finger	−0.0126	0.2639	0.5531	1.0377	1.9541	4.8345	0.9996
Lip	−0.0001	0.1981	0.4966	0.9943	2.0075	5.1500	0.9999
Variance	Finger	2.9 × 10^−5^	3.8 × 10^−5^	5.7 × 10^−5^	8.5 × 10^−5^	1.1 × 10^−4^	7.6 × 10^−4^	
Lip	2.8 × 10^−6^	2.8 × 10^−4^	1.8 × 10^−3^	2.3 × 10^−5^	3.4 × 10^−4^	2.1 × 10^−3^

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
