# Peer review of "Wireless Sensing of Lower Lip and Thumb-Index Finger ‘Ramp-and-Hold’ Isometric Force Dynamics in a Small Cohort of Unilateral MCA Stroke: Discussion of Preliminary Findings"

_sensors, 2020, doi:10.3390/s20041221_

Round 1
Reviewer 1 Report
It is an interesting framework but I feel that the paper is not suitable for this journal.
I believe that a full disclosure of the hardware device is needed to appeal the audience of this journal.
Some other concerns:
1) the result presentation should be revised, the ones presented in Figures 7 and 8 show a better separation between stroke patients and controls, as they are easier to understand for non-medical background readers should be presented first.
2) Without a proper disclosure of the measurement tool used it is difficult to put some of the presented results in the appropriate context. As example the data presented in figure 5 shows large variability around small force values and some of the ranges i.e. the ones around 2 N (LL) are quite comparable. What is the uncertainty of these measurements (hardware) and how this affect the fine variations between control and stroke patient, in other words is the tool fine enough to resolve with a good precision and accuracy the variation??
3) is the tool clinically relevant? In other words is there a practical use for this tool in stroke rehabilitation?
Reviewer 2 Report
The article appears too wordy and very hard to read. Effectively it revolves around a minimal cohort of just 7 MCA stroke patients, which also appear very dishomogeneous in terms of clinical history and extent of the ischaemic territory. I am therefore not entirely sure that this article is adding significantly to the current literature.
I would suggest to slim the result section and use bullet points to summarise the main findings.
I would kindly ask to justify how the size of controls was calculated.
I would request the authors to modify the title as follows: 'Wireless Sensing of Lower Lip and Thumb-index Finger ‘Ramp-and-Hold’ Isometric Force Dynamics in a small cohort of Unilateral MCA Stroke: Discussion of Preliminary Findings'.
Reviewer 3 Report
Thanks for the opportunity to review this interesting manuscript. This is a novel study with nice methology as well as presenting a well-written manuscript. Nevertheless, the sample is not balanced and there is a lack of a sample size calculation. I suggest authors to balance the sample with seven and seven subjects due to you state that "Seven chronic (mean = 70 months post-stroke) hemiparetic male MCA stroke survivors (mean = 90 46.7, SD = 20.1, range = 23-67 years), and 25 neurotypical adult males (mean = 30.0, SD = 14.9, range = 91 19-65 years)." This could improve your study conclusions. In addition, please specify the study design as well as followed guidelines. A sample size calculation would be useful. If you can not provide a sample size calculation, I suggest authors to state this study as a pilot study, as well as including this issue in limitations section.
Round 2
Reviewer 1 Report
I can see that the manuscript has improved and many of my concerns have been addressed. I believe that tho make the manuscript of interest to the readers of this journal, the use for rehab described between lines 513-522 needs to be expanded and described better also in the introduction. Its limitations based on the statistic separation of the trials between the 2 groups for each of the class should be restated in the conclusion section.
Reviewer 3 Report
Thanks for the authors' response. There is one issue which remains without response according to my prior review:
"In addition, please specify the study design as well as followed guidelines". Please, revise it.
